# Floquet engineering non-equilibrium steady states

Alberto Castro[1,2]* and Shunsuke A. Sato[3,4]

**1** Institute for Biocomputation and Physics of Complex Systems, University of Zaragoza, 50018 Zaragoza (Spain)
**2** ARAID Foundation, 50018 Zaragoza (Spain)
**3** Center for Computational Sciences, University of Tsukuba, Tsukuba 305-8577, Japan
**4** Max Planck Institute for the Structure and Dynamics of Matter, Luruper Chaussee 149, 22761 Hamburg, Germany
* acastro@bifi.es

January 27, 2023

## Abstract

Non-equilibrium steady states are created when a periodically driven quantum system is also incoherently interacting with an environment – as it is the case in most realistic situations. The notion of *Floquet engineering* refers to the manipulation of the properties of systems under periodic perturbations. Although it more frequently refers to the coherent states of isolated systems (or to the transient phase for states that are weakly coupled to the environment), it may sometimes be of more interest to consider the final steady states that are reached after decoherence and dissipation take place. In this work, we demonstrate how those final states can be optimally tuned with respect to a given predefined metric, such as for example the maximization of the temporal average value of some observable, by using multicolor periodic perturbations. We show a computational framework that can be used for that purpose, and exemplify the concept using a simple model for the nitrogen-vacancy center in diamond: the goal in this case is to find the driving periodic magnetic field that maximizes a time-averaged spin component. We show that, for example, this technique permits to prepare states whose spin values are forbidden in thermal equilibrium at any temperature.

# 1 Introduction

Exploring novel materials in search of desired properties and functionalities is one of the most important tasks of material sciences and engineering, as it can significantly impact fundamental sciences and practical applications. For example, the conversion efficiency of solar cells has been significantly enhanced over the past several decades through the discovery of various types of materials [1–3]. Likewise, thanks to the exploration in a vast materials space, various superconducting materials have been found [4–7]. In addition to these examples, various materials explorations have been conducted toward the realization of desired material properties and functionalities in the equilibrium phase [8–10].

Recently, the exploration and design of material functionalities has been extended to the nonequilibrium phase of matter under the presence of optical or magnetic drivings. In the seminal work by Oka and Aoki [11], for example, the light-induced anomalous Hall effect in graphene has been theoretically studied in terms of the Floquet picture, suggesting the emergence of topological states of matter. Inspired by this work, various groups have investigated the emergence of new material properties under electromagnetic drivings. The design of material functionalities in the nonequilibrium phase has thus become a full new field of research, that is often called *Floquet engineering* [12–17].

In most theoretical works about Floquet engineering, the states of the target system have been investigated by considering the time-periodic solutions of the Schrödinger equation. However, real materials are surrounded by their environment, and those Floquet states, which are the time-periodic solutions of the Schrödinger equation, may decay quickly and not be relevant. In fact, recent theoretical and experimental studies suggest that the realization of the Floquet states can be significantly disturbed by their interaction with the environment [18–22]. For a practical description of such driven systems, a theory of open-quantum systems under periodic driving has to be considered. However, understanding such driven nonequilibrium phases is significantly more difficult.

Recently, we have demonstrated [17] an approach to Floquet engineering based on the use of quantum optimal control theory (QOCT) [23–27]: the idea was to allow for multicolor periodic driving, rather than the monochromatic ones that are normally assumed, and to use the tools of QOCT to find the amplitudes of the various frequency components that optimize a given target property of the system – in that work, the goal was to modify at will the (pseudo) band structure of graphene.

However, that work also ignored the effect of the environment, and therefore, the found optimal states would only live in a transient prethermalized phase. To realize the Floquet control of material properties and functionalities in systems more tightly coupled to an environment, going beyond the conventional Floquet analysis for isolated systems, we extend here that previous concept of Floquet engineering to open-quantum systems. For this purpose, we first discuss how to apply optimal control theory for nonequilibrium steady states of open-quantum systems under periodic driving, based on a quantum master equation. We then apply the introduced optimal-control procedure to a model of the NV center of diamond under periodic driving, demonstrating that, for example, driven open quantum systems under optimized fields may display exotic properties that are forbidden in the equilibrium phase. Although to our knowledge, no previous work has attempted the optimization of NESSs with respect to the external drivings, a related work [28] has recently demonstrated the use of automatic differentiation to optimize steady states with respect to internal system or bath parameters.

## 2 Method

In order to manipulate the nonequilibrium steady states, we solve the following optimization problem. Our first assumption is to consider, as master equation, a Lindblad-type equation [29, 30] with time-periodic external fields:

$$\dot{\rho}(t) = -i[H(t), \rho(t)] + \sum_{ij} \gamma_{ij} \left( V_{ij} \rho(t) V_{ij}^\dagger - \frac{1}{2}\{V_{ij}^\dagger V_{ij}, \rho(t)\} \right). \tag{1}$$

Here, the Hamiltonian $H(t + T) = H(t)$ is periodic with time period $T$. We consider it to be composed of a field-free and and a periodic perturbation part: $H(t) = H_0 + g(u, t)V$, where $g(u, t) = g(u, t + T)$ is some $T$-periodic real function parametrized by the set $u = u_1, \ldots, u_P$ – the *control parameters*. The incoherent part of the evolution is determined by the set of Lindblad operators $V_{ij}$, which we will assume in the following, without loss of generality, to be the transition operators $V_{ij} = |E_i\rangle\langle E_j|$, where $|E_i\rangle$ are the field free Hamiltonian eigenvectors.

   We should warn that the previous equation is not universally valid. In fact, the problem of deriving valid master equations for systems with time-dependent Hamiltonians is still an open research area. The equation of Lindblad can only be rigorously derived if the Hamiltonian is time independent – and even then, it rests on several additional conditions, most notably Markov's approximation. Various authors have tackled the problem of deriving master equations for driven systems [31–34]. In some circumstances, Lindblad-type equations with time-dependent Hamiltonians such as Eq. (1) are appropriate [35], and have been used for various purposes [36–38]. The previous equation is a simplified version of the so-called Floquet-Lindblad equation [39]. We will work with it as working hypothesis; furthermore, the optimization procedured described below can be easily generalized to more complex master equations.

   A Lindblad equation such as the one above can always be written as a linear equation in Liouville space:

$$\dot{\rho}(t) = \mathcal{L}(u, t)\rho(t), \tag{2}$$

where we now consider $\rho(t)$ to be in vectorized form, i.e it is a $N^2$-dimensional complex vector vector, where $N$ is the dimension of the underlying Hilbert space [40]. The Lindbladian $\mathcal{L}(u, t)$ is the $N^2 \times N^2$ dimensional operator that results of transforming Eq. (1) into this space. We split it as:

$$\mathcal{L}(u, t) = \mathcal{L}_0 + g(u, t)\mathcal{V}. \tag{3}$$

   Let us call $\rho_u(t)$ to the periodic solution (i.e. $\rho_u(0) = \rho_u(T)$) of Eq. (2) for a set of parameters $u$. This solution corresponds to a non-equilibrium steady-state (NESS). Note that, in principle, there could be more than one steady state, but we will consider here that it is unique. We then consider the time-average function

$$F(\rho) = \frac{1}{T} \int_0^T dt\, \tilde{A}(\rho(t)), \tag{4}$$

for some function of density matrices $\tilde{A}$ – in practice, this will typically be the expectation value of some operator $A$: $\tilde{A}(\rho) = \text{Tr}[A\rho]$. The problem that we attempt to solve is the optimization of function:

$$G(u) = F(\rho_u), \tag{5}$$

subject perhaps to some constraint on the parameters $u$.

   Such class of optimization problems for time-dependent processes that can be controlled by the manipulation of external handles is the object of (quantum, in this case) optimal control theory (QOCT). Any function optimization algorithm requires a method for the computation

of the function; in addition, many efficient algorithms will also require a method for the computation of its gradient. Computing the function $G$ essentially amounts to obtaining the NESS. In the following, we will show one possible way to do this, and also derive one expression for the gradient. Note that since

$$G(u) = \frac{1}{T} \int_0^T dt \, \text{Tr}[A\rho_u(t)], \tag{6}$$

the gradient components may then be computed as:

$$\frac{\partial G}{\partial u_k} = \frac{1}{T} \int_0^T dt \, \text{Tr}[A\frac{\partial \rho_u}{\partial u_k}(t)], \tag{7}$$

and therefore the problem in fact amounts to finding some procedure to compute the derivatives $\frac{\partial \rho_u}{\partial u_k}$.

Let us first rewrite Eq. (2) elementwise:

$$\dot{\rho}_\alpha(t) = \sum_\beta \mathcal{L}_{\alpha\beta}(u, t)\rho_\beta(t) \tag{8}$$

and consider the Fourier transform of these objects:

$$\rho_\alpha(t) = \sum_n \rho_{\alpha,n} e^{i\omega_n t}, \tag{9}$$

$$\rho_{\alpha,n} = \frac{1}{T} \int_0^T dt \, e^{-i\omega_n t} \rho_\alpha(t), \tag{10}$$

$$\mathcal{L}_{\alpha\beta}(u, t) = \sum_n \mathcal{L}_{\alpha\beta,n}(u) e^{i\omega_n t}, \tag{11}$$

$$\mathcal{L}_{\alpha\beta,n}(u) = \frac{1}{T} \int_0^T dt \, e^{-i\omega_n t} \mathcal{L}_{\alpha\beta}(u, t), \tag{12}$$

where $\omega_n = \frac{2\pi}{T}n$, $n = 0, 1, \ldots, N-1$. In the frequency domain, the Lindblad equation, Eq. (2), can then be rewritten as [1]:

$$\sum_\beta \sum_{m=0}^{N-1} \left[ \mathcal{L}_{\alpha\beta,n-m}(u) - i\delta_{nm}\delta_{\alpha\beta}\omega_m \right] \rho_{\beta,m} = 0. \tag{13}$$

And, by further defining the following operator

$$\overline{\mathcal{L}}_{\alpha n,\beta m}(u) = \mathcal{L}_{\alpha\beta,n-m}(u) - i\delta_{nm}\delta_{\alpha\beta}\omega_m, \tag{14}$$

we finally rewrite Eq. (2) as:

$$\sum_\beta \sum_{m=0}^{N-1} \overline{\mathcal{L}}_{\alpha n,\beta m}(u)\rho_{\beta,m} = 0. \tag{15}$$

---

[1]These equations are easily reached using the following two formulas:

$$\frac{1}{T} \int_0^T dt \, \dot{\rho}_\alpha(t) e^{-i\omega_n t} = i\omega_n \rho_{\alpha,n},$$

and

$$\frac{1}{T} \int_0^T dt \, \mathcal{L}_{\alpha\beta}(u, t)\rho_\beta(t) e^{-i\omega_n t} = \sum_{n=0}^{N-1} \mathcal{L}_{\alpha\beta,n-m}(u)\rho_{\beta,m}.$$

This is a linear homogeneous equation; the solution (the nullspace or kernel, assuming that it has dimension one), will be the periodic solution that we are after, the NESS [2]. We now need some procedure to find $\frac{\partial \rho}{\partial u_k}$. Taking variations of Eq. (15) with respect to the parameters $u$, we get:

$$\overline{\mathcal{L}}(u)\frac{\partial \rho}{\partial u_k}(u) = -\frac{\partial \overline{\mathcal{L}}}{\partial u_m}(u)\rho_u. \tag{16}$$

This is a linear equation that would provide $\frac{\partial \rho_u}{\partial u_k}$. However, note that since $\overline{\mathcal{L}}(u)$ has a non-empty kernel (given precisely by $\rho_u$), it cannot be solved straightforwardly. In fact, it does not have a unique solution: If $x$ is a solution of

$$\overline{\mathcal{L}}(u)x = -\frac{\partial \overline{\mathcal{L}}}{\partial u_m}(u)\rho(u), \tag{17}$$

$x + \mu\rho_u$ is also a solution for any $\mu$. To remove this arbitrariness, we impose the normalization condition, $\text{Tr}\rho_u = 1$ for any $u$, and therefore:

$$\text{Tr}\frac{\partial \rho_u}{\partial u_k} = 0. \tag{18}$$

To find $\frac{\partial \rho_u}{\partial u_k}$ in practice, we may then take the following two steps: First, we compute a solution of the linear equation, Eq. (17), with the least-squares method, by imposing that the solution $x_0$ is perpendicular to the kernel, i.e.: $x_0^\dagger \cdot \rho_u = 0$. Then, we update the solution with the condition, Eq. (18). The required solution is obtained as:

$$\frac{\partial \rho_u}{\partial u_k} = x_0 - (\text{Tr}x_0)\rho_u. \tag{19}$$

Once we have $\frac{\partial \rho_u}{\partial u_k}$, we can evaluate the gradient in Eq. (7). Armed with this procedure to compute this gradient, one can perform the optimization of function $G(u)$ with many efficient algorithms.

These methods have been implemented in the qocttools code [41], publicly available, and all the necessary scripts and data necessary to replicate the following results are also available upon request from the authors.

## 3 Results

In the following, we will use the previous equations with the following model of the NV center of diamond [36, 42]:

$$H(u,t) = H_0 + V(u,t), \tag{20}$$
$$H_0 = -B_s S_z + N_z S_z^2 + N_{xy}(S_x^2 - S_y^2), \tag{21}$$
$$V(u,t) = -g_x(t)B_d S_x - g_y(t)B_d S_y. \tag{22}$$

The model definition must be completed with the definition of the dissipative part: we take $\gamma_{ij} = \gamma e^{-\beta E_i}/(e^{-\beta E_i} + e^{-\beta E_j})$ and $\gamma_{ii} = 0$, where $\beta = 1/(k_B T)$ is the inverse of the temperature, and $\gamma$ is a rate constant [3]. The reason for choosing this model is the work of Ikeda *et al.* [36], who studied the NESSs of this system under circularly polarized light ($g_x(t) = \cos(\omega t); g_y(t) = \sin(\omega t)$).

---

[2]Other procedures could be used to compute the NESS, sometimes also called "asymptotic Floquet states", such as for example simply propagating the equation for a long time, as the system should decay to the steady state.

[3]Notice that this dissipation model ensures the detailed balance condition, $\gamma_{ij}e^{-\beta E_j} = \gamma_{ji}e^{-\beta E_i}$.

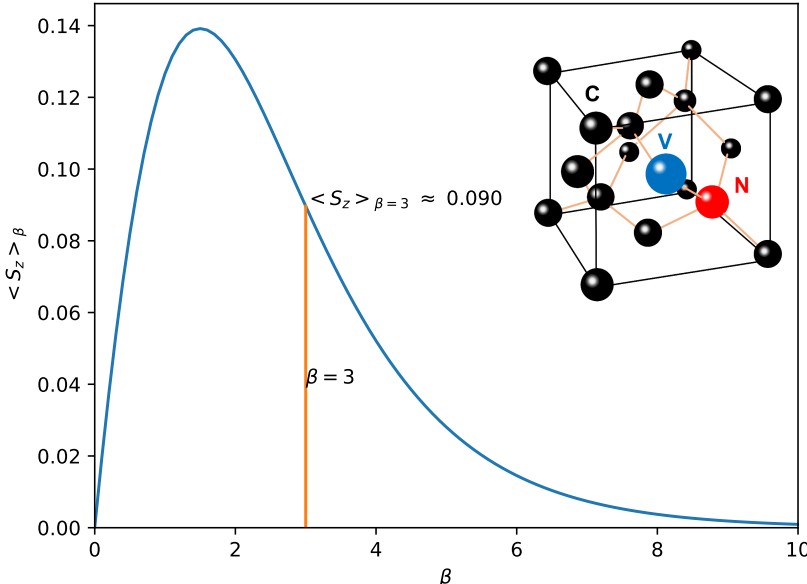

Figure 1: Thermal average of $S_z$, as a function of the inverse temperature $\beta = \frac{1}{k_B T}$. The value at $\beta = 3$, used in the text for the rest of the calculations, is singled out. *Inset*: structure of the Nitrogen vacancy defect in diamond.

In that work, the high-frequency approximation was used in order to derive simplified expressions for the NESS. Here, the goal would be to parametrize functions $g_x = g_x(u, t)$ and $g_y = g_y(u, t)$, and find the parameters $u$ that result in a NESS that maximizes the time-averaged value of some observable (for example, $S_z$).

Following Ikeda *et al.* [36], we set the units of the model by fixing $N_z = 1$; the rest of the parameters of the model are then given by: $N_{xy} = 0.05, B_s = 0.3, B_d = 0.1, \gamma = 0.2$ (see [42] for a review on the NV diamond centers, this and other models, and the typical values that these constants may take).

First, let us consider the field-free value of $S_z$; the thermal average of $S_z$, $\langle S_z \rangle_\beta$, is shown in Fig. 1 as a function of the inverse temperature $\beta$. One can see how at zero temperature ($\beta \to \infty$), $\langle S_z \rangle_\beta \to 0$, reflecting the fact that the ground-state value of $S_z$ is also zero: $\langle \psi_0 | S_z | \psi_0 \rangle = 0$. As the temperature increases, the population of the first excited state grows, and therefore the thermal average of $S_z$ also grows, since $\langle \psi_1 | S_z | \psi_1 \rangle \approx 1$. However, if the temperature is increased further, the population of the second excited state also starts to grow, and the thermal average starts to decrease, as $\langle \psi_2 | S_z | \psi_2 \rangle \approx -1$. In the limit of infinite temperature ($\beta \to 0$), the thermal average approaches zero again, as that limit involves an equally populated ensemble of all three states. Note then that a *thermal control* of $S_z$, i.e. the manipulation of the value of $S_z$ via a variation of the temperature, is limited to the range $0 < \langle S_z \rangle_\beta < 0.14$.

However, as we will show, if a periodic perturbation is added, this range can be enlarged, and one may reach NESSs with larger or smaller values of the (time averaged) $S_z$. In the following, let us fix $\beta = 3$, and seek for the drivings that are capable of producing those NESSs. The first step is to set a parametrized form for the time-dependent functions $g_x$ and

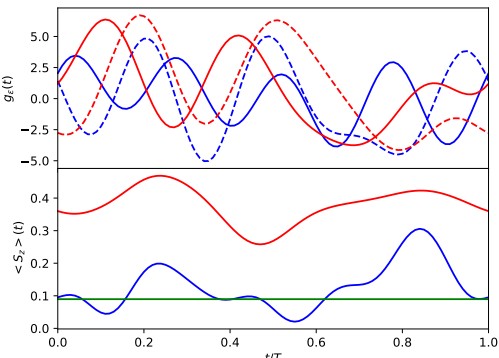 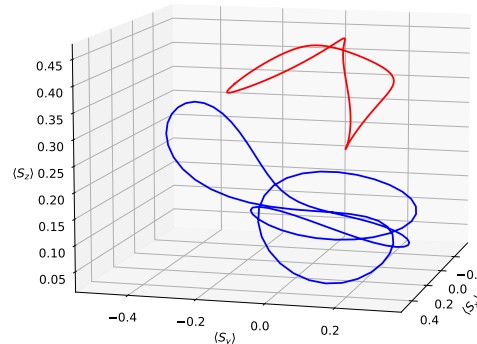

Figure 2: *Left, top:* Optimized (red) and initial guess (blue) temporal shapes of the time-dependent magnetic fields $g_x$ (solid) and $g_y$ (dashed). *Left, bottom:* Evolution of $\langle S_z \rangle$ when using the initial guess (blue) and the optimal fields (red). The green line represent the thermal average at $\beta = 3$. *Right:* Trajectories of the spin vector $\langle \vec{S}(t) \rangle$ during one period $T$, for the initial guess (blue) and optimized (red) perturbations.

$g_y$ used in Eq. (22); the simplest choice is to use Fourier expansions:

$$
g_x(u,t) = u_0 + \sum_{n=1}^{M} [u_{2n}\cos(\omega_n t) + u_{2n-1}\sin(\omega_n t)] , \tag{23}
$$

$$
g_y(u,t) = u_{2M+1} + \sum_{n=1}^{M} [u_{2M+1+2n}\cos(\omega_n t) + u_{2M+2n}\sin(\omega_n t)] .
$$

The control parameters are therefore the Fourier coefficients of the temporal shape of the two magnetic fields, $u_0, \ldots u_{4M+1}$. The index $M$ determines the cutoff frequency $\omega_M$, whereas all the Fourier frequencies are $\omega_n = n\omega_0$ for $n = 1, \ldots, M$. A choice must then be made on the fundamental frequency $\omega_0$, which is of course related to the period that we choose for the external field $\omega_0 = \frac{2\pi}{T}$. In this work, we have chosen $\omega_0 = 0.5\, N_z$, and $M = 4$, such that the cutoff frequency is $\omega_M = 2.0\, N_z$. By defining the control functions in this parametrized manner, we effectively constrain the final solution to a given domain of validity – in this case setting a maximum frequency. This would be consistent with any experimental realization of this concept, as in practice the time-dependent magnetic fields would also be constrained in frequencies due to technological limitations.

The optimization of function (6) may then be started using any gradient-based algorithm – the one that we have used for these calculations is the Sequential Least-Squares Quadratic Programming (SLSQP) algorithm [43] as implemented in the NLOPT library [44]. Note that we have not performed an unconstrained maximization for all possible values of parameters $u_j$, but we have added a constraint on the amplitudes of each frequency component:

$$
|u_j| \leq \kappa \quad \text{for any } j. \tag{24}
$$

Such a constraint would also be present in an experiment. The chosen algorithm permits to include this constraint.

Fig. 2 shows the results of one optimization; in this case the amplitudes were constrained using $\kappa = 4.0$. The optimization is started with random fields (shown in the left, top panel, with blue lines), and then proceeds iteratively until the fields that optimize the temporal average of $S_z$ are found (shown in the left, top panel, with red lines). In the left, bottom panel, the evolutions in time of $S_z$ are shown, once again for the initial guess and for the optimized

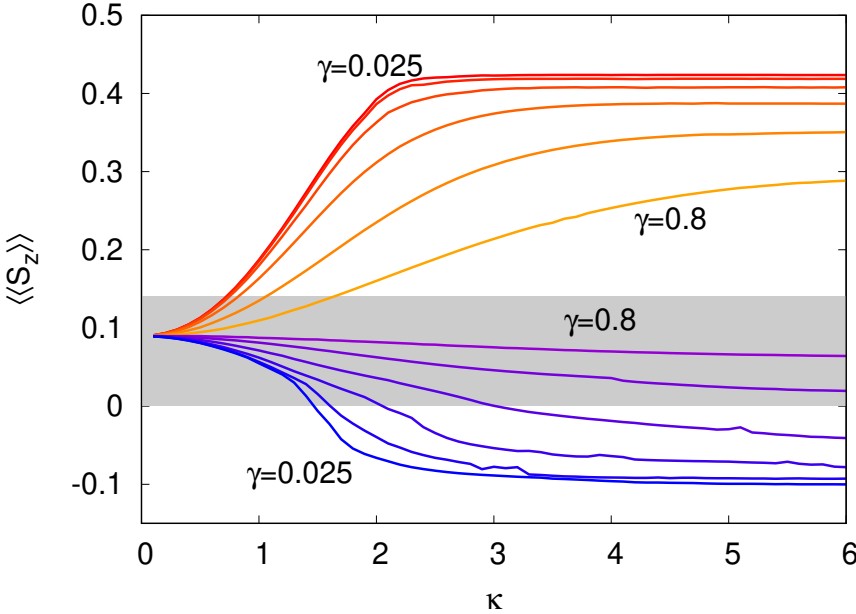

Figure 3: Maximized (red-orange) and minimized (blue-violet) values of the time-averaged $S_z$ expectation value, $\langle\langle S_z\rangle\rangle$, as a function of the amplitude bound $\kappa$. The various curves correspond to different values of the rate constant $\gamma$, which are doubled from $\gamma = 0.025$ to $\gamma = 0.8$. The shaded region marks the only allowed values of $S_z$ in thermal equilibrium (thus, for example $\langle S_z\rangle_\beta > 0$).

case. It can be seen how the optimized fields lead to significantly higher values of $S_z$ – both with respect to the initial random fields, and with respect to the thermal value (shown as a straight green line in the plot). In fact, the time-averaged value of $S_z$ achieved in this way ($\approx 0.38$) is higher than the maximum that can be achieved in equilibrium phase by modifyng the temperature ($\approx 0.14$, as discussed above). The right part of Fig. 2 shows the full spin vector $\langle\vec{S}(t)\rangle$ evolving in time during one Floquet period, both for the initial (blue) and optimized (red) cases.

The final optimized value of function $G$ (i.e. of the time averaged value of $S_z$) obviously depends on how we constrain the periodic functions. For example, on the bound $\kappa$ that we set on the amplitudes. Fig. 3 shows the optimal value obtained as a function of that bound (red curves), for various values of the dissipation constant $\gamma$. Obviously, if the bound is set to a very small value, the presence of the periodic field barely modifies the thermal average (of around 0.09, for the chosen temperature value, $\beta = 3$). However, if the bound is relaxed to higher values, the average can be significantly increased, up to a saturation value that depends on $\gamma$: the higher the $\gamma$, the lower the value of the optimized $\langle\langle S_z\rangle\rangle$. This can be understood physically, as a faster dissipation drives with more strength the system towards its thermal equilibrium state. Finally, we have attempted to *minimize* the time average of $S_z$, wondering whether one can engineer states with the in principle forbidden negative spin values. In Fig. 3 we display the obtained optimal values, also as a function of the amplitude bound (red curves). It may be seen how, if sufficiently big amplitudes are allowed, one may actually obtain negative values – which are forbidden in thermal equilibrium, as it can be seen in Fig. 1.

## 4 Conclusions

We have developed an optimal control scheme for the nonequilibrium steady states of open quantum systems under time-periodic drivings, aiming to control the properties of matter in nonequilibrium phases. We derived an expression for the gradient vectors of physical observables in NESSs with respect to the parameters of the external periodic fields, and we employed these derived gradient vectors for the optimization of observables of the diamond NV center under external periodic magnetic fields. We confirmed that the time-averaged value of the spin component, $S_z$, can be controled with the proposed optimal control sheme. Furthermore, we demonstrated that this technique can be used to find "exotic" NESSs, such as states that display properties that are forbidden in equilibrium phases: As shown in Fig. 3, the $z$-spin component of the optimized NESS can be outside the range of values allowed in equilibrium – for example, it may be negative, which is impossible at any temperature.

Having established an optimal control scheme for NESSs under periodic driving, the field parameters can be added as novel degrees of freedom for material explorations aimed to endow the materials with desired properties and functionalities. This extends the concept of material exploration, from equilibrium to nonequilibrium situations. Because the present optimization scheme is based on the steady state solutions of a master equation, such as Lindblad's equation [Eq. (1)], the relaxation and dissipation effects are naturally included in the optimization procedure. Hence, the engineering of material properties based on the proposed scheme can be seen as an extension of the more common Floquet engineering usually based on the steady solutions of the time-dependent Schrödinger equation without taking into account the relaxation and dissipation effects. The optimal control of NESSs proposed in this work shows how the difficulties of Floquet engineering due to the relaxation and dissipation effects can be overcome, and the natural inclusion of these effects opens a path to the control of material properties with experimentally realizable fields.

## Acknowledgements

**Funding information**     AC acknowledges support from Grant PID2021-123251NB-I00 funded by MCIN/AEI/10.13039/501100011033. SAS acknowledges the support from JSPS KAKENHI Grant Numbers JP20K14382.

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
