# Peer review of "Floquet engineering non-equilibrium steady states"

_SciPost Physics_

## Round 1 · Referee Report · Anonymous (Referee 1) · 2023-3-11

Strengths

New numerical method.
Clear application.

Weaknesses

Lack of comparison with other methods.

Report

Report attached

Requested changes

Indicated in the report.

Attachment

  • validity: high
  • significance: good
  • originality: good
  • clarity: good
  • formatting: good
  • grammar: excellent

Author:  Alberto Castro  on 2023-04-04  [id 3539]

(in reply to Report 1 on 2023-03-11)

(see attached file)

Attachment:

author-response-1.pdf

---

## Round 1 · Referee Report · Anonymous (Referee 2) · 2023-3-21

Strengths

1- concise description of methodology 2- conceptual advance in the field of optimal control of driven-dissipative steady states

Weaknesses

1- computational costs are not discussed 2- limitations of the method are not discussed

Report

Generally, this is an interesting technical piece of work, and can be published in SciPost Physics. The other reviewer has already mentioned a few points of potential improvement that should be addressed. Besides those, it would be good if the authors could also mention the scope of this method -- what are requirements on the model for this method to work efficiently? The chosen example of NV center in diamond is a small model system with limited Hilbert space. Is it clear that the method can be expanded to extended systems -- solids are mentioned in the motivation? If so, what are the restrictions on those systems? E.g., does it only work efficiently for essentially noninteracting problems? Once a more in-depth discussion of these issues is added, I do recommend publication in SciPost Physics.

Requested changes

1- discuss costs and limitations

  • validity: high
  • significance: high
  • originality: high
  • clarity: high
  • formatting: excellent
  • grammar: excellent

Author:  Alberto Castro  on 2023-04-04  [id 3540]

(in reply to Report 2 on 2023-03-21)

We thank the referee for the recommendation to publish, and for the suggestion, that we find very appropriate. We have added a discussion about this point to the manuscript at the end of section 2

---

## Round 1 · Referee Report · Anonymous (Referee 3) · 2023-3-28

Report

This paper offers a way for designing Floquet driving protocols as an optimization problem, maximizing an observable in the nonequilibrium steady states (NESSs). Taking advantage of the simple structure of the Lindblad equation, the authors obtain gradients for the density matrix in terms of driving parameters, which allows the optimization of the NESSs.

Roughly speaking, this work is a direct combination of Ref. 28 and Ref. 36. Reference 28 discusses the optimization problem without Floquet drivings, whereas Ref. 36 discusses the NESS without optimization. In this view, the current title, "Floquet engineering non-equilibrium steady states," is inappropriate and too general since this concept was already proposed, at least in Ref. 36.

Besides, the paragraph "However, ..." in the Introduction is referenced only by Ref. 28, and the authors completely ignore all the relevant works on Floquet engineering in open systems, including the NESS, even stating, "we extend here that previous concept of Floquet engineering to open-quantum systems." The authors' contribution is not to extend Floquet engineering to open systems but to develop an optimization technique for (previously proposed) Floquet engineering of NESSs. I strongly recommend that the authors read a recent review article, arXiv:2203.16358, and revise the Introduction and title so as to highlight their contribution correctly.

Given that this work is a direct combination of Refs. 28 and 36, I do not find it a groundbreaking discovery or a breakthrough that is required in the acceptance criteria. So I cannot recommend publication in this journal. Since the technical results seem valid and interesting, I recommend publication in a non-flagship journal like SciPost Physics core after making appropriate revisions.

Requested changes

1- Modify the title more specifically. 2- Revise the Introduction by adding more references on Floquet engineering in open systems.

  • validity: high
  • significance: high
  • originality: good
  • clarity: high
  • formatting: excellent
  • grammar: excellent

Author:  Alberto Castro  on 2023-04-04  [id 3541]

(in reply to Report 3 on 2023-03-28)

(see attachment)

Attachment:

author-response-3.pdf

---

## Editorial Decision

resubmitted